# Nursing-Intense Health Education Intervention for Persons with Type 2 Diabetes: A Quasi-Experimental Study

**DOI:** 10.3390/healthcare9070832

**Published:** 2021-07-01

**Authors:** María Begoña Martos-Cabrera, José Luis Gómez-Urquiza, Guillermo Cañadas-González, José Luis Romero-Bejar, Nora Suleiman-Martos, Guillermo Arturo Cañadas-De la Fuente, Luis Albendín-García

**Affiliations:** 1Neonatal Intensive Care Unit, University Hospital San Cecilio, Avenida del Conocimiento, 18016 Granada, Spain; begomartos90@gmail.com; 2Faculty of Health Sciences, University of Granada, Avenida de la Ilustración, 60, 18016 Granada, Spain; jlgurquiza@ugr.es (J.L.G.-U.); norasm@ugr.es (N.S.-M.); gacf@ugr.es (G.A.C.-D.l.F.); 3Support Device South Area of Cordoba, Andalusian Health Service, Av. Góngora, 9B, Cabra, 14940 Córdoba, Spain; benaldoctor@hotmail.com; 4Department of Statistics and Operational Research, University of Granada, 18071 Granada, Spain; 5Casería de Montijo Health Center, Granada Metropolitan District, Andalusian Health Service, Calle Virgen de la Consolación, 12, 18015 Granada, Spain; lualbgar1979@ugr.es

**Keywords:** clinical trial, diabetes mellitus type 2, HbA1c, health education, primary care

## Abstract

Type 2 diabetes mellitus (DM2) is a highly prevalent disease, the progression of which depends on high blood glucose levels, which are reflected in the level of glycosylated haemoglobin (HbA1c). Appropriate health education equips patients with the knowledge and skills to control their glucose and HbA1c levels to avoid long-term complications. This study was set up to compare the results of an intensive (360 min) educational intervention to improve HbA1c parameters in patients with DM2 with those of a usual 90 min intervention. For this purpose, healthcare personnel led a quasi-experimental study of 249 diabetics: 171 in the control group, and 78 in the intervention group. In the control group, the mean HbA1c value decreased from 6.97 to 6.75, while in intervention group it fell from 8.97 to 8.06. The before and after mean difference between both groups was compared with a Wilcoxon test, and the results statistically significant (W = 4530; *p* < 0.001), indicating a higher reduction of HbA1c in the intervention group. We concluded that the intensive health education provided by nurses during the consultation helped improve HBA1c levels in persons with DM2.

## 1. Introduction

The American Diabetes Association defines diabetes mellitus (DM) as a group of metabolic diseases characterised by hyperglycaemia resulting from defects in insulin secretion, insulin action, or both. Chronic hyperglycaemia of diabetes is associated with long-term damage [1,2].

According to the aetiology of chronic hyperglycaemia, diabetes mellitus is classified as type 1 (DM1), type 2 (DM2) or gestational (DMG). DM1 is characterised by the absence of insulin secretion by an autoimmune process [1,2,3,4]. DM2 arises from a defect in the effect of insulin in the tissues or in their resistance to it, and the majority of patients are overweight [1,3,5,6].

Between 1980 and 2014, the prevalence of DM2 rose from 4.7 to 8.5%, and premature mortality due to DM increased by 5%, making it the seventh cause of death in 2016. It is estimated that by 2030 366 million patients worldwide will have DM. In Europe, the prevalence is expected to increase by 16% between 2017 and 2045 [7,8]. In 2016, 13.8% of the Spanish adult population had DM2 according to the di@betes.es study (conducted in Spain) [7]. Given the age and sex composition of the population, an incidence of 11.6 cases per 1000 inhabitants per year was estimated, which corresponded to 3.7 cases/1000 inhabitants/year of known diabetes symptoms and 7.9 cases/1000 inhabitants/year of unknown symptoms [7,8].

Between 90 and 95% of patients with DM are classified as type 2. The majority are between 50 and 60 years and overweight. The disease may be triggered by various environmental factors [1,9,10,11,12], most of which are modifiable, meaning that DM2 is potentially preventable. Accordingly, by addressing these factors the disease can be controlled both to improve the patient’s quality of life and prevent secondary pathologies arising from inadequate blood glucose management [3,7,12].

Among the many risk factors for DM2 is being overweight or obese, with the latter contributing the most to the development of insulin resistance [12]. Family history, hypertension, dyslipidaemia and age are also relevant, with those over 45 years having a higher probability of presenting the disease [7]. Moreover, a personal history of DMG increases the possibility of developing DM2 by up to 10 times compared to pregnant women with normal blood sugar levels [13]. Finally, an inadequate diet or a sedentary lifestyle also increase the risk of DM2 [1,3,5,7,14].

The usual treatment for DM2 is focused on ensuring that blood glucose remains within appropriate bounds. This parameter is measured by the glycosylated haemoglobin (HbA1c) test, which provides recent data on fluctuations in blood glucose levels (usually, the last three months). The target value for good glycaemic control in patients with DM2 is HbA1c <7% [15,16].

To achieve the recommended HbA1c level, patients with DM2 are given treatment to maintain or achieve normoglycaemia. This treatment is both pharmacological, based on subcutaneous insulin and oral medication, and non-pharmacological through education, nutrition and physical exercise. Information about these approaches is provided during doctor–patient consultations, and support material, such as mobile applications and games, can be used to facilitate learning and management [17,18,19,20,21].

Determining and conducting the interventions for controlling glycaemia in these patients may be complex. The process requires knowledge of the disease and the ability to perform the prescribed treatment. Therefore, health education is of vital importance for patients with DM2. In fact, the risk of mortality is inversely related to the patient’s understanding of the disease [22].

Previous studies with Health Education interventions have focused mainly on adherence to good nutritional habits, such as the Mediterranean diet [23], which improves HbA1c. Other positive results stem from interventions like personalized nutrition [24], adequate knowledge about counting carbohydrates [25] and consulting nutritionists [26].

Other studies have focused on interventions led by nursing staff [27], therapy relating to acceptance and of the disease and a commitment to lessening it [28] and self-care support [29] by reporting health improvements.

Providing information and treatment procedure training during consultations, according to the patient’s understanding, can lower HbA1c levels, which can also prevent pathologies such as retinopathy, nephropathy and neuropathy [5,15,30].

In this context, the patient is not only informed about the disease, but is also shown the technique and frequency of blood glucose measurement, medication administration, the importance of suitable nutrition, and the benefits of physical exercise. Therefore, it is vital that this consultation be conducted by trained physicians and nurses since all interventions must be integrated for global treatment [31].

In short, the aim of our study is to determine whether the provision of an intensive health education programme by healthcare personnel improves HbA1c parameters in patients with DM2.

## 2. Materials and Method

### 2.1. Design and Participants

From 1 July 2019 to 31 January 2020, a quasi-experimental study with a control group was performed to compare basic and intensive systematic interventions with the usual care offered at primary care clinics of the Andalusian Health Service (Spain). The reference value taken was the prevalence of DM2 as determined by the epidemiological analysis of the comprehensive diabetes care plan conducted in Andalusia prior to 2019 [30,31].

Our analysis of the results was based on a conservative measurement of each case, and the expectation that the non-systematic intervention control group would obtain only limited benefit, compared with significantly greater improvements for the two intervention groups [15,22,32]. Patients with higher HbA1c levels were included in the intervention group intentionally because, according to previous studies, they could acquire early glycaemic control and delay the introduction of oral antidiabetic drugs, according to the II Comprehensive Diabetes Plan of Andalusia [33].

The patient’s condition was classified as DM2, if he or she was older than 30 and presented any of the following criteria: overweight or obesity, insidious onset, absence of ketonuria/ketonaemia, family history of DM2, or a personal history of DMG; otherwise, the condition was classified under “other types of diabetes” [1,3,5,13].

The study population was 291 patients, but the final sample was composed of 249 with DM2 (171 in the control group and 78 in the intervention group) because some patients refused to participate. They were recruited consecutively according to their order of receiving attention at the clinic until the previously established sample size was obtained (calculated according to the total number of diabetic patients in the health area). These patients were referred from nursing or medical consultations by qualified healthcare professionals at each location. Patients with very high HbA1c levels (>7%) were assigned to intervention group, and the rest were assigned to the control group [3,15]. Thus, the selection process was intentional.

### 2.2. Inclusion Criteria

All participants were diagnosed with DM2 at primary health care centres in Andalusia and were stratified by age and type of diabetes. All participants were treated in the Granada-Metropolitan Health District for the duration of the study, received the requisite HbA1c tests, and agreed to remain in the study for approximately six months, attending all scheduled control visits and receiving the appropriate interventions.

### 2.3. Interventions Performed and Study Groups

The interventions for the control and the intervention group were performed during individual sessions by the same two nurses, who had been working with people with diabetes for more than 10 years and who also had postgraduate training in diabetes management. The characteristics of the usual care for the control group and the intensive education intervention for the intervention group are listed below.

Control group (patients with HbA1c levels <7%):-15 min scheduled consultation plus group support, every 15 days for 3 months-Education for health in diabetes: diet and exercise.-Nursing care plan.-Examination of sensitivity to pressure and vibration in both feet.-HbA1c analysis after 6 months.

Intervention group (patients with HbA1c levels >7%):
-30 min scheduled consultation plus group support, every 7 days for 3 months-Education for health in diabetes: improved self-care techniques in relation to the complications of poorly controlled diabetes.-Diet and exercise.-Nursing care plan: assessment and evolution of nursing objectives.-Determination of the ankle-brachial index.-Detection of vascular complications by anamnesis and examination.-Screening for diabetic retinopathy: digital retinography.-HbA1c analysis after 6 months

### 2.4. Variables and Data Collection

Sociodemographic data (age and sex) and health outcomes (body mass index, presence of hypertension, retinopathy, neuropathy and diabetic foot risk) were collected at baseline at the first education sessions. Also, 6 months after the intervention, HbA1c was collected through a blood analysis.

### 2.5. Statistical Analysis

SPSS version 20 was used for all statistical analyses. Mean and standard deviation were used for description of continuous variables and frequencies for qualitative variables. A Wilcoxon test for related samples was used for a before-and-after means comparison because the variables did not follow a normal distribution. The mean reduction between both groups was compared with the Wilcoxon test.

## 3. Results

The control group sample was *n* = 171 and the intervention group’s was *n* = 78. The control mean age was 65.32 (SD 13.45) and the intervention mean age was 63.93 (SD 11.18). Regarding body mass index, the control mean was 29.95 (SD 5.35) and intervention mean was 30.57 (SD 5.49). In the control group, the mean HbA1c value pre intervention was 6.97 (SD 1.36) and for the intervention group, 8.97 (SD 1.60). The other characteristics at baseline are shown in Table 1.

After the intervention, the mean HbA1c value in the control group was 6.75 (SD 0.84) and 8.06 (SD 1.40). in the intervention group. The analysis showed that in both groups there was a significant reduction in HbA1c levels (Table 2).

The pre- and post-intervention mean difference in both groups was compared with the Wilcoxon test, and the results were statistically significant (W = 4530; *p* < 0.001), indicating a higher reduction of HbA1c in the intervention group.

## 4. Discussion

The effective monitoring and management of DM2 is of the utmost importance. It has been estimated that for every 1% by which HbA1c is reduced in patients with poorly controlled DM2, the probability of premature death falls by 21% and that of microvascular complications by 37%. The duration of DM2 is also associated with premature death from cardiovascular disease [34,35,36].

Therefore, health education is vital and must be in accordance with each patient’s education and understanding of the disease and its different concepts. In this respect, a low level of education has been related to the insufficient consumption of whole grains and thus to an increased risk of DM2-related complications [34,37].

In view of these considerations, any programme aimed at instructing DM2 patients on self-care and independence should take into account the patient’s understanding of the question. Although most studies of such initiatives have failed to observe a statistically significant improvement in HbA1c, there have been improvements at the psychosocial level [38]. Some patients may follow the recommendations of health professionals for a short time, but in the absence of a long-term follow-up and proper management, the improvement will not persist [39,40]. Similar outcomes have been observed among patients with other pathologies, which underlines the view that a health education programme requires follow-up to maintain its initial benefits [41].

Studies have shown that intensive, appropriate health education for DM2 patients leads to improved HbA1c levels, and that a multidisciplinary team, in which nurses personalise the intervention and facilitate communication [42,43] is essential for its success. Nurses also foster the patient’s ability to self-manage the disease, and this not only promotes effective decision making to achieve clinical targets, such as improving blood sugar levels, but it also contributes to reducing disease-related distress [42,44,45,46]. Previous research in this field has also shown that the role played by nurses as health educators enhances patients’ quality of life in areas such as adhering to the Mediterranean diet, reducing consumption of carbohydrates, and improving blood sugar values [47,48].

In view of the great importance of the nurse–patient relationship, consideration should be given to maintaining it over a longer period via electronic means or telephone calls. This approach would reinforce patients’ knowledge of their health status (for example, the level of HbA1c), enhance decision-making in this respect, and at the same time, provide cost–benefit advantages for the healthcare system [49]. Indeed, action protocols have been proposed based on this understanding to reduce the risk of complications in patients with poorly controlled DM2 and elevated HbA1c (>7%) [50].

Other studies have shown that increasing the frequency of patients’ visits to the nursing consultation improves their quality of life and reinforces lifestyle changes. However, to achieve an effective relationship of trust and to provide the necessary help, it must be established at an early stage. Another crucial question is the patient’s motivation, which is directly associated with the likelihood of HbA1c levels being improved and the enhancement of personal well-being [51,52].

However, opinions are divided. Some researchers have failed to confirm the above considerations, in that statistically significant results were not achieved in patients with elevated levels of HbA1c (>8%), possibly because clear objectives were not established and agreed upon with the patient during which health education consultations with the nurse, [53]. Another inconvenience is that some patients had not achieved long-term improvement by the end of the study period and the intensive follow-up was terminated [54].

Among the ways in which health education may be undertaken, it has been suggested that follow-ups might be improved by “peer training”, whereby other patients with DM2 and well acquainted with the condition provide help and advice; however, the assistance and support of qualified nursing staff would still be needed, for example, to correct false beliefs [55,56,57].

Family-focused interventions can also help patients improve self-care and alleviate anxiety. Nevertheless, without the support of appropriate healthcare professionals, it is unlikely to improve HbA1c levels on its own [58].

The inclusion of cognitive-behavioural therapy as an additional nursing technique may benefit health outcomes by, for example, improving adherence to non-pharmacological treatments [59].

The inclusion of different interventions is important to delay or avoid the inclusion of oral antidiabetics. According to the II Comprehensive Diabetes Plan of Andalusia, oral antidiabetic drugs should be prescribed if after 3–6 months there is no improvement in HbA1c with adequate adherence to exercise and healthy nutrition [31]. Educational programs, according to previous studies, can improve HbA1c levels and delay the inclusion of oral antidiabetic drugs or insulin in the treatment of patients with DM2 [60,61]. Other studies showed that oral antidiabetic drugs and insulin improve HbA1c levels but significantly worsen the quality of life [62]. In addition, patients with this type of treatment require a more specific health education [63,64]. Thus, patients must also implement other resources to face different events throughout their disease and be able to maintain metabolic balance [65].

In summary, we believe a simple medical consultation is not enough for patients with DM2 and that a more complex intervention is required to reduce HbA1c levels and achieve lasting benefit.

### Limitations

The study population was drawn from patients who attended primary-care consultations. This circumstance may have resulted in their presenting a greater willingness to comply with the medical and lifestyle indications provided, thus biasing the results. Furthermore, due to the quasi-experimental design and the intentional sampling, baseline levels of HbA1c and other pathologies were different between groups. Thus, the results must be accepted with caution.

Another limitation that the period allowed for observing changes in health outcomes may have been insufficient in some cases. Although six months is a substantial period for assessing changes in HbA1c levels, an intervention lasting 12 months, providing greater intensity and content, might have obtained even better results. As the patient’s knowledge and skills are enhanced, health education procedures should be modified accordingly.

## 5. Conclusions

The findings of the study showed that intense health education interventions by nursing staff can make a real contribution to patients with DM2, as their levels of HbA1c and adherence to good health habits clearly improved. On the other hand, consideration should be given to extending the follow-up provided to maintain long-term benefits. Moreover, appropriate staff training is required to ensure that patient education groups are effectively managed and to personalise care from a holistic perspective, so that patients with DM2 may consolidate and expand their acquired knowledge and habits.

## Figures and Tables

**Table 1 healthcare-09-00832-t001:** Characteristics at baseline.

Variable		Control Group (*n* = 171)	Intervention Group 2 (*n* = 78)	*p*
Sex	Male	55%	61.4%	*p* < 0.05
	Female	45%	38.6%	
Hypertension	Yes	63.3%	59.3%	*p* > 0.05
	No	36.84%	40.7%	
Retinopathy	Yes	17.5%	22.2%	*p* < 0.05
	No	82.5%	77.8%	
Neuropathy	Yes	28.1%	67.9%	*p* < 0.05
	No	71.9%	32.1%	
Diabetic foot	Normal	66.7%	60.5%	*p* < 0.05
	Risk	25.7%	30.9%	
	High risk	7.6%	8.6%	

*p* = *p* value comparison at baseline.

**Table 2 healthcare-09-00832-t002:** HbA1c Means pre and post intervention.

Group	Pre-Intervention	Post-Intervention	Mean Difference, Pre and Post	W	*p*
Control	6.97 (SD 1.36)	6.75 (SD 0.84)	0.22	1872	<0.001
Intervention	8.97 (SD 1.60)	8.06 (SD 1.40)	0.91	2704	<0.001

df = Degrees of freedom; *p* = *p* value; SD = Standard deviation; W = Wilcoxon test.

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
