# Peer review of "Nursing-Intense Health Education Intervention for Persons with Type 2 Diabetes: A Quasi-Experimental Study"

_healthcare, 2021, doi:10.3390/healthcare9070832_

Round 1

Reviewer 1 Report

The current article highlights the effect of intense health education intervention on type-2 diabetes management and found a positive correlation. The following points need to be addressed for a better presentation of the findings-

1- In the introduction, there is sufficient mention of diabetes cause and treatment but there is no or very limited information regarding previous studies investigating the effect of health education on diabetic outcomes. It is important to note previous research and shortfalls to appreciate the current research.

2- This study uses a quasi-experimental approach. It needs to be mentioned why the randomized model was not selected.

3- Using non-randomized participant selection creates unequal participants in both groups and different baseline HbA1c levels where the control group has lower levels and the experimental group has higher levels. Similarly, as mentioned in table 1, the control group has significantly lower levels of neuropathy (28.1%) compared to the treatment group (67.9%). It's important to discuss the effect of these baseline differences on final outcome.

Author Response

Dear reviewer,

Thank you for reviewing the manuscript and for your comments to improve it. Please find below the response to each comment highlighted in yellow and also the changes in the manuscript.

Kind regards

REVIEWER 1

The current article highlights the effect of intense health education intervention on type-2 diabetes management and found a positive correlation. The following points need to be addressed for a better presentation of the findings-

  • In the introduction, there is sufficient mention of diabetes cause and treatment but there is no or very limited information regarding previous studies investigating the effect of health education on diabetic outcomes. It is important to note previous research and shortfalls to appreciate the current research.

We have included in the introduction more information and references to other studies investigating the effect of health education on diabetic outcomes.

  • This study uses a quasi-experimental approach. It needs to be mentioned why the randomized model was not selected.

We have included the information on why we have selected a quasi-experimental approach in the methods section.

  • Using non-randomized participant selection creates unequal participants in both groups and different baseline HbA1c levels where the control group has lower levels and the experimental group has higher levels. Similarly, as mentioned in table 1, the control group has significantly lower levels of neuropathy (28.1%) compared to the treatment group (67.9%). It's important to discuss the effect of these baseline differences on final outcome.

We have included information about this problem in the limitations section. Because the intervention group had higher HbA1c levels they had more neuropathy.

Reviewer 2 Report

This is a very interesting but not surprising study. It is very well established that DM education improves outcomes, so it is not a novel study. However, this message is extremely important for DM care worldwide and still underappreciated by many providers or teams. Hence, another great example and success story of DM education will be a powerful and very helpful tool for diabetes care. Almost 1% Hba1c drop is even greater than some of the pharmacological interventions, which should have been discussed in the article. I would create a table showing side by side Hba1c reducing effects of anti-diabetics and this study, which would demonstrate the message much better. They might also discuss the comparison between other major DM education intervention results and their study, differences, advantages and weaknesses of their study, which would provide higher scientific quality.

Additionally :

Table 1. Characteristics at baseline. needs to be presented with p values because there are differences between the control and intervention groups, such as HTN presence etc. It is not acceptable to show the percentage only. 

Overall, I support this publication with some revisions and improvements.

Author Response

Dear reviewer,

Thank you for reviewing the manuscript and for your comments to improve it. Please find below the response to each comment highlighted in yellow and also the changes in the manuscript.

Kind regards

REVIEWER 2

This is a very interesting but not surprising study. It is very well established that DM education improves outcomes, so it is not a novel study. However, this message is extremely important for DM care worldwide and still underappreciated by many providers or teams. Hence, another great example and success story of DM education will be a powerful and very helpful tool for diabetes care. Almost 1% Hba1c drop is even greater than some of the pharmacological interventions, which should have been discussed in the article. I would create a table showing side by side Hba1c reducing effects of anti-diabetics and this study, which would demonstrate the message much better. They might also discuss the comparison between other major DM education intervention results and their study, differences, advantages and weaknesses of their study, which would provide higher scientific quality.

In the discussion we talk about the importance of health education to delay or to avoid the introduction of oral anti-diabetics. We have not included a table comparing both effects because it is not usual to include tables in the discussion.

Additionally : Table 1. Characteristics at baseline. needs to be presented with p values because there are differences between the control and intervention groups, such as HTN presence etc. It is not acceptable to show the percentage only.  Overall, I support this publication with some revisions and improvements.

We have included the p value in table one and also information about the baseline differences in the limitations sections.